# Analysis of a new negevirus-like sequence from *Bemisia tabaci* unveils a potential new taxon linking nelorpi- and centiviruses

Diego F. Quito-Avila[1,2]*, Edison Reyes-Proaño[3], Gerardo Armijos-Capa[4], Ricardo I. Alcalá Briseño[5], Robert Alvarez[6], Francisco F. Flores[7,8]*

**1** Centro de Investigaciones Biotecnologicas del Ecuador, CIBE, Escuela Superior Politécnica del Litoral, ESPOL, Campus Gustavo Galindo, Guayaquil, Ecuador, **2** Facultad de Ciencias de la Vida, Escuela Superior Politécnica del Litoral, ESPOL, Guayaquil, Ecuador, **3** Department of Entomology, Plant Pathology and Nematology, University of Idaho, Moscow, ID, United States of America, **4** Facultad de Ciencias Exactas, Departamento de Química, Instituto de Investigaciones Fisicoquímicas Teóricas y Aplicadas (INIFTA), Universidad Nacional de La Plata, CCT La Plata-CONICET, La Plata, Argentina, **5** Department of Plant Pathology, Oregon State University, Corvallis, OR, United States of America, **6** Department of Plant Pathology, University of Minnesota, St Paul, MN, United States of America, **7** Departamento de Ciencias de la Vida y la Agricultura, Universidad de las Fuerzas Armadas-ESPE, Sangolquí, Pichincha, Ecuador, **8** Facultad de Ciencias de la Ingeniería e Industrias, Centro de Investigación de Alimentos, CIAL, Universidad -UTE, Quito, Pichincha, Ecuador

* dquito@espol.edu.ec (DFQ-A); fjflores2@espe.edu.ec (FFF)

**Data Availability Statement:** The data underlying the results presented in the study are available

## Abstract

This study presents the complete genome sequence of a novel nege-like virus identified in whiteflies (*Bemisia tabaci* MEAM1), provisionally designated as whitefly negevirus 1 (WfNgV1). The virus possesses a single-stranded RNA genome comprising 11,848 nucleotides, organized into four open reading frames (ORFs). These ORFs encode the putative RNA-dependent-RNA-polymerase (RdRp, ORF 1), a glycoprotein (ORF 2), a structural protein with homology to those in the SP24 family, (ORF 3), and a protein of unknown function (ORF 4). Phylogenetic analysis focusing on RdRp and SP24 amino acid sequences revealed a close relationship between WfNgV1 and Bemisia tabaci negevirus 1, a negevirus sequence recently discovered in whiteflies from Israel. Both viruses form a clade sharing a most recent common ancestor with the proposed nelorpivirus and centivirus taxa. The putative glycoprotein from ORF 2 and SP24 (ORF 3) of WfNgV1 exhibit the characteristic topologies previously reported for negevirus counterparts. This marks the first reported negevirus-like sequence from whiteflies in the Americas.

## Introduction

Negeviruses comprise a group of arthropod-infecting viruses with monopartite single-stranded (+) RNA genomes. The generic name derives from the prototype Negev virus (strain E0239) discovered in *Anopheles coustani* from the Negev Desert in Israel [1]. Negeviruses have been found all over the world in several bloodsucking arthropod species in the genera *Anopheles*, *Culex*, *Lutzomyia*, *Aedes*, *Armigeres*, *Mansonia*, *Wyeomyia*, *Trichoprosopon*, *Coquillettidia*,

from NCBI GenBank under accession number PP036886.

**Funding:** The author(s) received no specific funding for this work.

**Competing interests:** The authors have declared that no competing interests exist.

*Psorophora*, *Toxorhychites*, *Ochlerotatus*, *Uranotaeneia*, and *Deinocerites* [1–11]. In recent years, however, the spectrum of natural hosts for nege-like viruses has expanded to encompass agriculturally significant insects like aphids, thrips, spider mites, and leafhoppers [12–16].

The genomes of negeviruses, spanning approximately 9.5 kb, are structured into three open reading frames (ORFs) and possess poly (A) tails. ORF 1 encodes the presumed viral replicase, a protein of around 260 kDa that encompasses methyltransferase (MTR), helicase (HEL), and RNA-dependent-RNA-polymerase (RdRp) domains. ORF 2 encodes a hypothetical glycoprotein (G) of about 46 kDa, while ORF 3 codes for a putative virion membrane protein of approximately 24 kDa, also known as structural protein of 24 kDa (SP24, Conserved Domain Database PF16504) [6,8,17,18]. Phylogenetic analyses based on RdRp sequences from negeviruses and nege-like viruses (collectively referred to as negevirids) have delineated two distinct, well-defined sister clades, termed nelorpivirus (derived from Negevirus, Loreto, and Piura viruses) and sandewavirus (named after Santana, Dezidougou, and Wallerfield viruses) [4,6,19]. Notably, a discernible evolutionary connection has been established between negevirids and plant-infecting viruses belonging to the family *Kitaviridae*, as evidenced by analyses involving both RdRp and SP24, the latter also identified in chroparaviruses, an insect-infecting virus taxon distantly related to nege/kitavirids [1,7,13,18,20,21]. Therefore, newly identified nege/kitavirids offer valuable insights into comprehending the evolutionary trajectories of these viruses, including their cross-kingdom transmissions and adaptive processes.

This investigation introduces, for the first time, the complete genomic sequence analysis of a novel negevirid discovered in the whitefly *Bemisia tabaci* MEAM1 (Gennadius), one of the most widely distributed plant-feeding insects [22]. Through comprehensive genome comparisons and phylogenetic analyses, we elucidate its evolutionary connections with other nege/kitavirids known to infect both arthropods and plants.

## Materials and methods

### Ethics statement

This research does not contain any essays involving human participants or animals. Field work and insect collections were carried out in strict accordance with the *Genetic Resource Access Permit* # MAE–DNB–CM–2018–0098 granted by the Department of Biodiversity of the Ecuadorean Ministry of the Environment.

### Virus source and high-throughput sequencing (HTS)

Ten adult whiteflies (*Bemisia tabaci*, MEAM1) were collected from papaya plants in a commercial orchard in Santa Elena, a coastal province of Ecuador. Specimens were immersed in 70% ethanol for transport and stored at -20˚C until processing. The ten whiteflies were pooled and subjected to total RNA extraction using the PureLink™ RNA Kit following manufacturer's instructions (Ambion, USA). After DNase treatment, the RNA was subjected to ribosomal RNA depletion and generation of cDNA library using Illumina's TruSeq RNA library prep Kit v2. The cDNA library was sequenced using Illumina's NovaSeq 6000 system generating 2x100 pair-end reads.

### Genome assembly and analysis

All the analyses of Illumina reads were conducted utilizing the tools integrated into Geneious Prime 2023. Following the process of pairing, duplicate reads were eliminated using dedupe, and low-quality reads were removed through BBDuk. *De novo* assembly was performed using SPAdes v. 3.15.5 [23]. Host associated contigs were first filtered out by performing BLASTn

using the megablast function (e-value 1e-10, word size 28). All contigs with $\geq$ 95% nt identity (query coverage $\geq$ 35%) with the Bemisia tabaci genome (GenBank assembly project: GCA_918797505.1) were removed. The homology of the remaining contigs was explored by BLASTx (e-value 0.01, word size 3) using the non-redundant (nr) protein database (as of February 2023). Contigs over 600 nucleotides in length with BLASTx hits to viral proteins showing amino acid identities $\geq$ 25% (e-value 0.01, query coverage $\geq$ 35%) were considered, for our purposes, "virus-related" sequences. All BLAST analyses were performed using the high-performance computing (HPC) cluster from Oregon State University's Center for Quantitative Life Sciences (CQLS).

The virus genome sequence assembled from HTS reads was used to design primers (S1 Table) to confirm the sequence in the original pooled RNA sample. Primers were designed in an overlapping fashion to cover the complete genome. Terminal regions were amplified using a second-generation RACE Kit (Roche, Germany) and specific primers designed close to the ends. Reverse transcription (RT) was done using SuperScript™ IV (Thermo Fisher Scientific, USA) and random primers following manufacturer's instructions. Polymerase chain reaction (PCR) was done using Platinum™ Superfi II DNA Taq polymerase, (Thermo Fisher Scientific, USA). PCR-amplified products of overlapping and terminal fragments were cloned using a pGEMT-easy Kit (Promega, USA) and sequenced by the Sanger method.

### Phylogenetic analysis

Evolutionary relationships of the new virus were investigated using the two most conserved proteins for nege/kitavirids, the RdRp and the putative SP24. Amino acid alignments and phylogenetics were performed using either protein but also using both proteins in a multi-locus type of analysis.

A total of 80 RdRp and 70 SP24 homologs were identified using the position-specific-iterated BLAST (PSI-BLAST) program [24] (https://BLAST.ncbi.nlm.nih.gov/BLAST.cgi?CMD=Web&PAGE=Proteins&PROGRAM=BLASTp&RUN_PSIBLAST=on) (accessed September, 2023) and downloaded from the National Center for Biotechnology Information (NCBI) GenBank. For most nege/kitavirids identified through PSI-BLAST both the RdRp and SP24 sequences were found. Exceptions include Bemisia tabaci nege-like virus 1, 2 and 3 (BtNeLV1-3), Aphis glycines nege-like virus 1 isolates AG1 and ABC1, Hubei virga-like virus 4, Frankliniella occidentalis associated negev-like virus 1–3, and Cordoba virus, for which no SP24 homologs were identified in GenBank during the analysis. Conversely, an amino acid sequence from *Drosophila melanogaster*, initially identified as a SP24 homolog by Quito-Avila et al. (2013) [25], was included in the analysis.

Amino acid multiple sequence alignments (MSA) were done using the MAFFT online server [26] and the transitive consistency score (TCS) [27] was applied for removing ambiguous alignment positions. The best amino acid substitution models (LG+G+I for both SP24, and RdRp) were determined using MEGA 11 [28]. The maximum-likelihood phylogenetic analyses for each protein and the concatenated alignment were performed with RAxML 8.2.12 [29] in the CIPRES Science Gateway [30]. The number of pseudo replicates used to determine the bootstrap support values was determined automatically in RAxML, according to the stopping criteria proposed by Pattengale et al. (2010) [31].

### Topology of putative structural proteins

The architecture of putative structural proteins, e.g. glycoprotein (ORF 2) and SP24 (ORF 3) was inferred using a systematic set of bioinformatic resources. Disordered regions were predicted using DISOPRED [32] from PSIPRED [33]. Hydrophobic regions were detected using

the Kyte-Doolittle hydropathy scale [34]. Transmembrane domains (TMs) were predicted using three different methods: the support vector machine-based MEMSAT algorithm [35], the hidden Markov model based TMHMM predictor server v. 2.0. [36] and the Constrained Consensus TOPology (CCTOP) prediction server v.1. [37].

## Three-dimensional (3D) modelling and structural comparison of SP4 homologs

The 3D structures of the putative SP24 from the newly sequenced virus and other seventy homologs were predicted using ColabFold v1.5.3 [38]. Briefly, input sequences in FASTA format were aligned against the UniRef100 and environmental sequences databases, resulting in the generation of two MSAs containing all detected sequences. For prediction of a single structure, a diversity-aware filter was applied to the MSAs to reduce redundancy and enhance diversity of sequences, followed by 3D prediction by AlphaFold2. Predicted local distance difference test (pLDDT) was applied as a confidence metric to rank single-structure predictions, where higher pLDDT values indicate more reliable and accurate predictions. For prediction of complex structures i.e. proteins with multiple interacting amino acid chains, the top hits within the same species were paired to resolve inter-chain contacts and facilitate the prediction of the entire complex's structure. ColabFold provides 3D structures in pdb format with an assessment of prediction quality including coverage sequences of MSA, pLDDT and predicted aligned error (PAE).

Upon obtaining the 3D-structures of seventy-one SP24 homologs, we conducted a comprehensive structural comparison using TM-align [39], which utilizes dynamic programming iterations to generate an optimized residue-to-residue alignment based on structural similarity. TM-score values range from 0 to 1, where values close to 1 indicate a high level of structural similarity, while lower scores indicate structural differences. Linear correlation between each pair of TM-scores was measured by Pearson Correlation Coefficient (PCC) [40] after TM-scores matrix transformation in Python 3 and represented by a heat map [41].

## Results

### HTS reads assembly

A total of 55.3 million paired-end sequencing reads were generated through the Illumina platform. Following the removal of duplicate and low-quality reads, *de novo* assembly produced 228,842 contigs ranging in length from 120 to 25,000 nt. Host associated contigs constituted 48.9% of the total, whereas only 0.05% (105 contigs) showed homology to different proteins encoded by RNA or DNA viruses associated with plants, fungi, insects, bacteria, or unclassified viruses (S1 Fig).

Within the set of 105 contigs, a particular 11.7 kb sequence displayed significant sequence homology to the RdRp found in various negevirus sequences documented in GenBank. The closest matches identified through BLASTx analysis were Bemisia tabaci negevirus 1 (BtNeV1, QWC36478.1) and BtNeLV1-3, exhibiting amino acid identity values of ~ 40% for a query coverage ~ 36% (e-value = 0.00001).

The assembly of the 11.7 kb contig was accomplished from 3,865 reads, with an average sequencing depth of 34X. A significantly higher read count was identified at the 3' end (Fig 1), as recently observed in Indomegoura nege-like virus 1 (INLV1) [14] and Astegopteryx formosana nege-like virus (AFNLV) [42]. This study focuses on the genomic analysis of this novel negevirus-like sequence, hereby designated as whitefly negevirus 1 (WfNgV1).

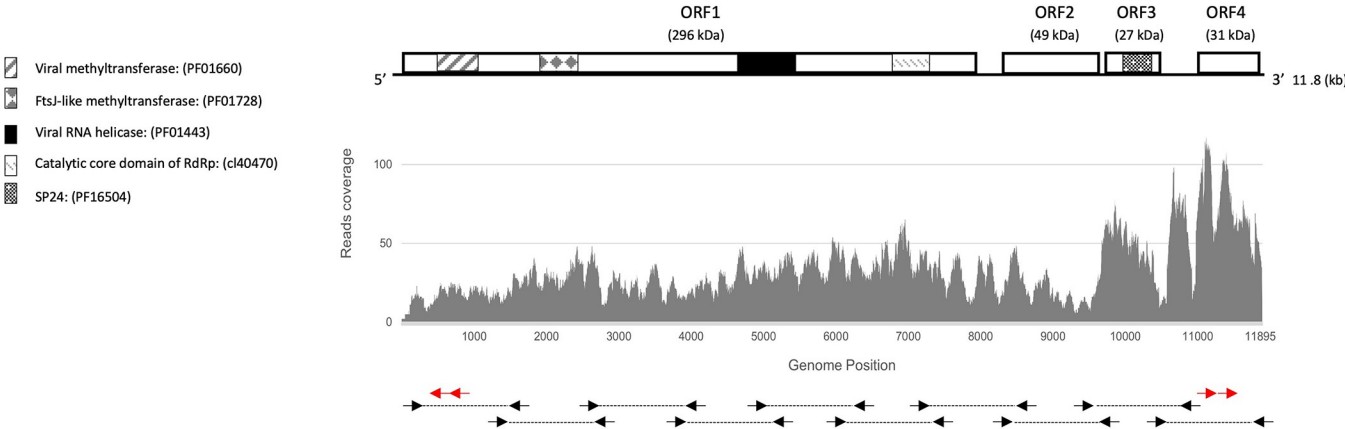

**Fig 1. Genome organization of whitefly negevirus 1 (WfNgV1) and read coverage from high throughput sequencing (HTS).** Predicted open reading frames (ORFs) with their corresponding molecular weight (kDa) are indicatated by rectangular boxes. Pattern-lined areas represent identified protein conserved domains as described in the legend. The lower panel shows the read coverage (y-axis) throughout the assembled genome. Arrows and dashed lines represent overlapping PCR amplified regions for Sanger sequencing (refer to S1 Table for primers sequence information). Red arrows positioned towards the 5'- and 3'- ends denote primer positions used for rapid amplification of cDNA ends (RACE).

## Genome organization and sequence homology

Sequencing of overlapping PCR products spanning the complete HTS-generated contig including the terminal regions revealed a 11,848 nt long genome (GenBank Acc. No. PP036886) with four predicted ORFs and poly (A) tail. ORF 1 (nt positions 108–7,883) encodes a 296.2 kDa protein with MTR, HEL, and a conserved catalytic RdRp domain found in kitavirids (cd23254). ORF 2 (nt positions 8,292–9,587) encodes a putative 49.4 kDa protein sharing ~20% amino acid identity with negevirus glycoproteins. ORF 3 (nt positions 9,629–10,357) codes for a 27.3 kDa protein with SP24 (pfam16504) conserved domain, showing ~30% amino acid identity with nege- and kitavirus counterparts, and ORF 4 (nucleotide positions 10,923–11,747) encodes a 31 kDa putative protein of unknown function (Fig 1). Table 1 presents a comparison between the main features of hypothetical proteins encoded by WfNgV1 and those encoded by its closest relative, BtNeV1.

## The putative RdRp of WfNgV1 shows motifs typically found in counterparts from nege/kita viruses

Studies have demonstrated that the RdRp of nege/kitavirids shares three conserved motifs with the RdRp of other single-stranded RNA viruses, specifically, motif A [DX(4–5)D], motif

**Table 1. Comparison between the putative proteins encoded by whitefly negevirus 1 (WfNgV1) and its closest relative Bemisia tabaci negevirus 1 (BtNeV1).** Molecular weight is expressed in kilodaltons (kDa).

| Hypothetical proteins | Protein features (length and molecular weight) | | | | | | Protein pairwise identity and accession numbers | | |
| --- | --- | --- | --- | --- | --- | --- | --- | --- | --- |
| | Nucleotide | | Amino acid | | kDa | | | | |
| | WfNgV1 | BtNeV1 | WfNgV1 | BtNeV1 | WfNgV1 | BtNeV1 | % | WfNgV1 | BtNeV1 |
| ORF 1 (Polymerase) | 7,776 | 3,089* | 2,591 | 1,028* | 296.2 | 118.6* | 42 | WRT26033 | QWC36478 |
| ORF 2 (Glycoprotein) | 1,296 | 1,242 | 431 | 413 | 49.4 | 48.5 | 22 | WRT26034 | QWC36479 |
| ORF 3 (Structural protein of 24 kDa, SP24) | 729 | 726 | 242 | 241 | 27.3 | 23.3 | 35 | WRT26035 | QWC36480 |
| ORF 4 (Unknown) | 825 | 765 | 274 | 254 | 30.9 | 29.3 | 24 | WRT26036 | QWC36479.1 |

* Partial sequence. It misses the 5' end of ORF1.

B [GX(2–3)TX(3)N], and motif C (GDD), arranged in either the sequence A-B-C or the permuted order C-A-B [3]. The RdRp of WfNgV1 and its closest relative BtNeV1, exhibits the A-B-C arrangement, with 47 and 12 amino acids between adjacent motifs. Comparative analysis across representative homologs from nege- and kitaviruses revealed additional conservation in the proximity or within the motifs, resulting in an expanded motif A, $^{2249}$DxxKYDKSQ$^{2257}$. Similarly, motif B displays five highly conserved amino acids, presenting the expanded motif $^{2305}$RxSGDxxTxxxNT$^{2317}$ in WfNgV1. It is noteworthy that BtNeLV1-3 (QWC36482, QWC36484, and QWC36485), display the C-A-B configuration of RdRp motifs (Fig 2A).

## ORF 2 deduced 49 kDa protein in WfNgV1 exhibits motifs shared among chropara- and negevirus glycoproteins

Kuchibhatla et al. (2014) studied and hypothesized the glycoprotein nature of the deduced product of ORF2 from the chronic bee paralysis virus (CBPV, the type member of the chroparavirus taxon) and negeviruses except those corresponding to the Sandewavirus clade. Although a low amino acid identity was observed across these homologs, a conserved region comprising approximately 50 amino acids was identified with four highly conserved cysteine residues predicted to form disulfide bridges [18]. We analyzed WfNgV1 putative glycoprotein (ORF2 product) and identified such a conserved region, indicating its evolutionary relationship, albeit with low amino acid identities (23.3% to 31.7%), with closest counterparts such as BtNeV1 (QWC36479) and Negev virus (BAR91506). The expected conserved cysteine residues were located in positions 70, 84, 90, and 114 of WfNgV1 glycoprotein. Additional conservation for residues valine (V), leucine (L), and proline (P), was observed across predicted glycoproteins of Aphis glycines virus 3 (ApGlV3, ASH89119) and Wuhan house centipede virus 1 (WHCV-1, BBV14742), among others (Fig 2B). Similar to glycoprotein homologs in CBPV and negeviruses, two putative transmembrane regions, TM1 (amino acid positions: 372 to 394) and TM2 (amino acid positions: 406–428) were predicted in the hypothetical WfNgV1 glycoprotein (not shown).

## WfNgV1 ORF 3 putative SP24 has transmembrane features

The comparison of amino acid sequences between the predicted SP24 of WfNgV1 and its nearest relatives (see the phylogenetic analysis section) displayed identities ranging from 19% to 34%. The most significant amino acid identity (34%) was noted with the corresponding sequence from BtNeV1, which also shares a similar length, featuring 241 amino acids compared to the 242 amino acids in the WfNgV1 SP24.

Analysis of the overall amino acid composition of WfNgV1 SP24 revealed a notable abundance of alanine (A, 10.3%), arginine (R, 9.1%), and L (8.3%), followed by P, isoleucine (I), and V each constituting 8%. The predicted molecular weight (M.W.) of this protein is 27.3 kDa. Its structural arrangement includes two disordered regions at the N- and C-termini, respectively, and an ordered portion featuring four highly hydrophobic motifs, indicative of transmembrane (TM) regions [18].

TM1 (aa positions 62–81) is delimited by conserved residues aspartic acid (D) and histidine (H), and is made up mostly of V (15%), phenylalanine (F, 15%), L (10%), serine (S, 10%), A (10%), and I (10%); whereas the region corresponding to predicted TM2 (aa positions 115–130) is preceded by a partially conserved WxxxNxxK motif with the core region mainly composed of I (25%), followed by P, glycine (G), A, F, and L, with 12.5% each. The interface between TM2 and TM3 is defined by a seven amino acid stretch, featuring an R located at the central interface segment, a characteristic observed in 70% of the homologs analyzed. The bulk of TM3 (aa positions 138–160) is made up mainly of I (17%) and V (13%), followed by P, F, A,

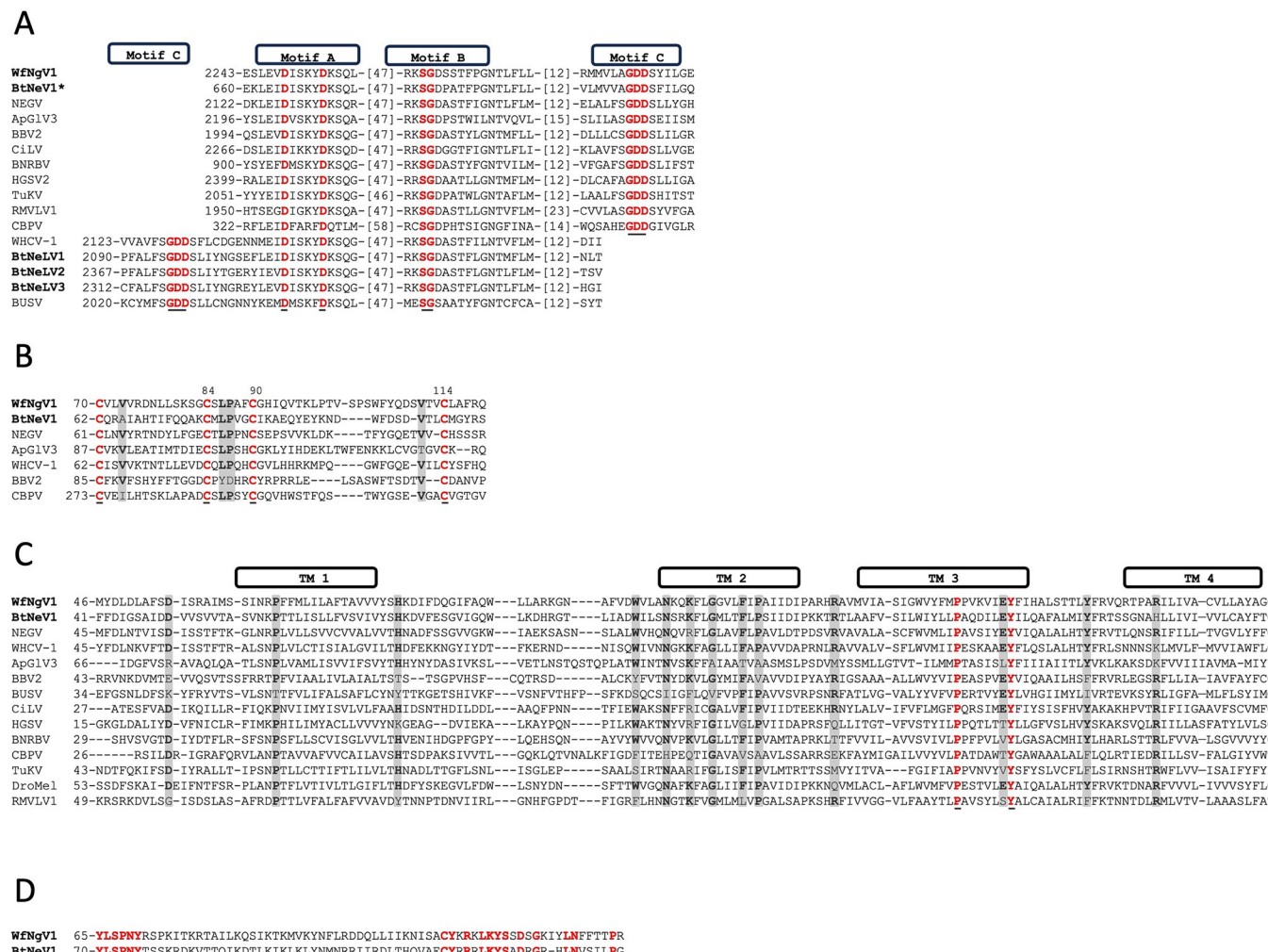

**Fig 2. Signature motifs in whitefly negevirus 1 (WfNgV1) hypothetical proteins and comparison across homologs form representative taxa.** A) Conserved motifs A, B, and C of the RNA-dependent-RNA-polymerase (RdRp). B) Conserved cysteine residues (WfNgV1 amino acid positions 70, 84, 90, and 114) characteristic of glycoproteins encoded by open reading frame (ORF) 2 of negevirids. Additional conserved residues are highlighted (shaded areas). C) Conserved residues (shaded areas) across hypothetical SP24 proteins from representative nege/kitavirids. Predicted transmembrane (TM) regions are indicated in boxes. D) Conserved residues corresponding to the hypothetical product of WtNgV1 ORF 4 and its closest relative. For all panels: Residues colored in red (also underlined) denote full conservation across analyzed homologs. Bolded acronyms represent those corresponding to whitefly nege/nege-like viruses. Asterisk denotes protein positions based on partial sequence. Numbers in brackets indicate number of non-conserved residues between conserved areas. Virus acronyms and taxa: Bemisia tabaci negevirus 1 (BtNeV1, unclassified), Negev virus (NEGV, Nelorpivirus), Aphis glycines virus 3 (ApGlV3, Aphiglyvirus), Beihai barnacle virus 2, (BBV2, unclassified), Citrus leprosis virus, (CiLV, Cilevirus), Blueberry necrotic ring blotch virus (BNRBV, Blunervirus), Hibiscus green spot virus 2 (HGSV2, Higrevirus), Tetranychus urticae kitavirus (TuKV, unclassified), Red mite virga-like virus 1 (RMVLV1, unclassified), Chronic bee paralysis virus (CBPV, Chroparavirus), Wuhan house centipede virus 1 (WHCV-1, Centivirus), Bemisia tabaci nege-like virus 1–3 (BtNeLV1-3, unclassified), Bustos virus (BUSV, Sandewavirus), DroMel (Drosophila melanogaster-associated SP24). Refer to S2 Table for accession numbers used in the alignments.

and tyrosine (Y), with 9% each. Interestingly, the motif **P**PVKVIE**Y** was identified, where the underlined proline and tyrosine residues constitute the most conserved residues across all homologs. The proline lays in the central region of TM3, whereas the tyrosine residue is found at the end of it. The ~12 aa long TM3—TM4 interface contains a tyrosine residue, which is present in 70% of all homologs. TM4 (aa positions 173–195), preceded by a conserved R residue, is mostly composed of A (22%) and L (17%), followed by Y, V, I, G, and threonine (T), with 9% each. The approximate location of TM regions in WfNgV1 SP24 and the conservation across representative homologs from nege/kitavirids are shown in (Fig 2C).

## WfNgV1 ORF4 encodes a hypothetical protein of unknown function

Despite extensive homology searches, no homologous sequences with predicted functions were identified for the hypothetical product of ORF4—a 31 kDa protein lacking discernible transmembrane motifs. However, a 24% amino acid identity was noted with its counterpart from BtNeV1, revealing two highly conserved motifs (Fig 2D). Hypothetical proteins from additional ORFs located downstream of the SP24 ORF in other negevirids did not align with those corresponding to WfNgV1 and BtNeV1.

## Phylogenetic analysis

The multiple sequence alignment of the RdRp was 4,658 amino acids in length and was reduced to 1,687 residues, with 23.7% pairwise identity, after eliminating ambiguous alignment positions. Phylogenetic inference based on the RdRp sequence placed WfNgV1 with its closest relative, BtNeV1, which was isolated from whiteflies in Israel. Both viruses shared a more recent ancestor with members of the nelorpivirus clade. Interestingly, three additional RdRp homologs from nege-like viruses found in whiteflies (BtNeLV1-3) collected in China formed a distinct clade positioned between nelorpiviruses (including WfNgV1 and BtNeV1), and those tentatively proposed as centiviruses [13] (Fig 3A). Removal of ambiguous alignment positions from SP24 multiple sequence alignment (422 aa in length) resulted in a 126 aa alignment with 26.8% pairwise identity. The analysis based on the SP24 sequence revealed that WfNgV1 and BtNeV1, form a clade alongside nelorpiviruses, centiviruses, and three unclassified viruses namely Beihai barnacle virus 2 (BBV2) [43], soybean thrips nege-like virus-1, and a Drosophila melanogaster-associated SP24 sequence [25]. However, this clade lacked robust support (bootstrap < 50%) (Fig 3B). In the concatenated tree combining RdRp and SP24 sequences, WfNgV1 and BtNeV1 were confidently grouped closer to nelorpiviruses in a well-supported clade (bootstrap 90%). Although the node corresponding to the divergence of nelorpiviruses and centiviruses exhibited low bootstrap support (51%), the node representing the most recent common ancestor for nelorpiviruses, centiviruses, and other nege-like homologs from B. tabaci was well-supported (bootstrap 100%). This suggests a clear separation of these viruses from other negeviruses, including those in the taxa Sandewavirus, Aphiglyvirus, and plant-infecting viruses in the family *Kitaviridae* (Bluner-, Cile-, and Higreviruses) (Fig 3C).

## Structural comparison of WfNgV1 SP24

The predicted 3D structure of WfNgV1 putative SP24 showed higher reliability between residues 30 and 190, where conserved positions suggest regions more likely to be accurately predicted during structure determination (Fig 4A and 4B). Within the prediction quality assessment, the selected model of WfNgV1 SP24 was based on the highest pLDDT values per residue position along the protein sequence. Most residues in the protein sequence had a pLDDT score above the threshold (70%) (Fig 4C), and low PAE values for residues within the 40–190 interval, indicating a higher confidence in the predicted local structure. Conversely, residues located at the N- and C- terminal regions showed higher PAE values, consistent with predicted disordered regions. Analysis of secondary structure revealed that 59.1% corresponds to helix structures, while the remaining fraction of the protein corresponds to coiled-coil structures, as determined by the dictionary of protein secondary structure (DSSP) method [44]. Structural analysis of seventy-one SP24 homologs from the TM-score matrix clustered homologs in a pattern reminiscent of our phylogenetic analysis, where SP24 homologs from WfNgV1, BtNeV1 and those from tentative members of the nelorpivirus and centivirus groups showed a closer evolutionary relationship (Fig 4D). A substantial difference was observed

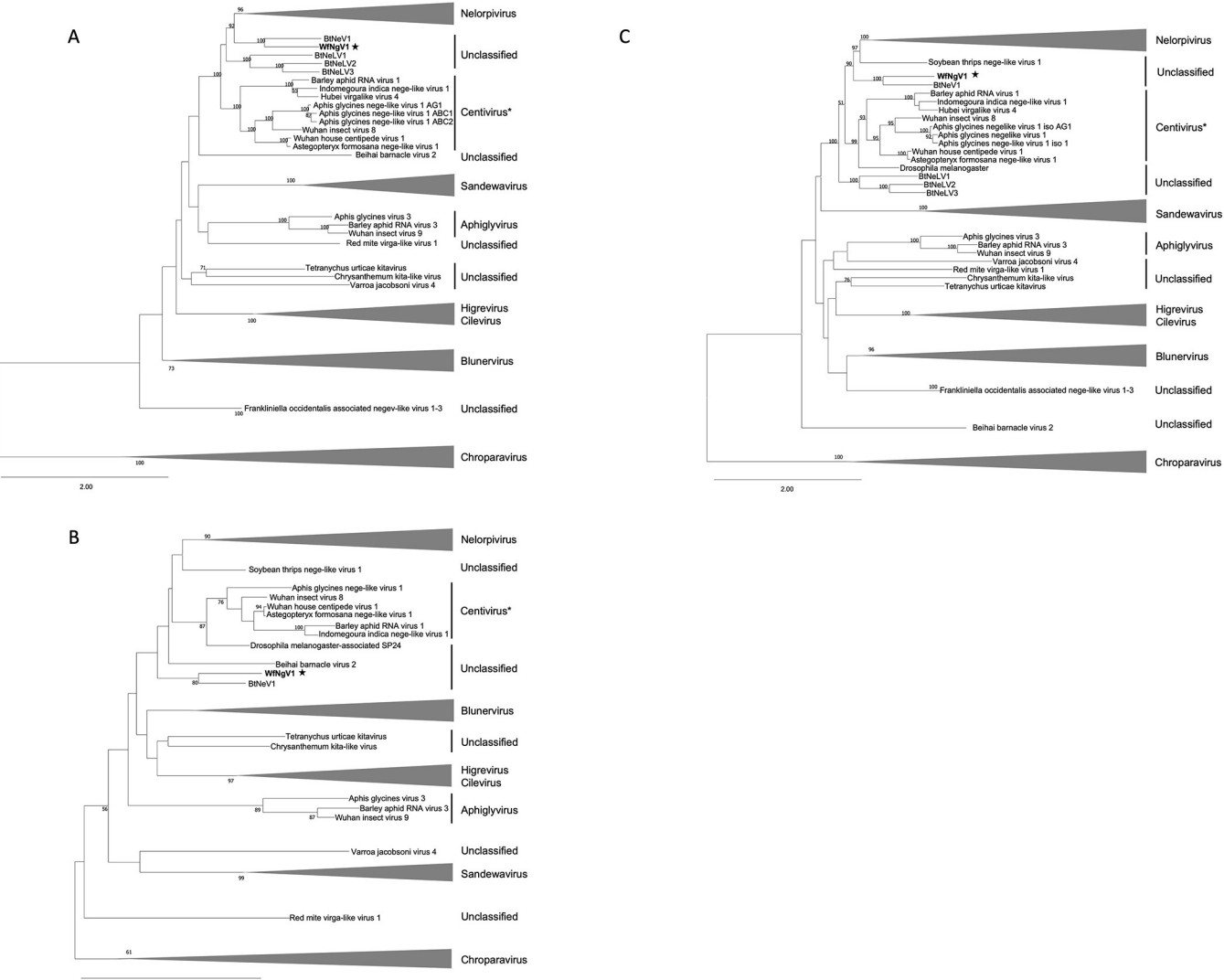

**Fig 3. Maximum likelihood phylogenetic trees depicting the relationships between whitefly negevirus 1 (WfNgV1) and representative nege/kita viruses sourced from GenBank.** The analysis utilized the amino acid sequences of the RNA-dependent-RNA-polymerase A), the structural protein of 24kDa, SP24 B), and the concatenation of both (C). The LG+G+I model served as the amino acid substitution model for all analyses. The tree is drawn to scale with branch lengths measured in the number of amino acid substitutions per site. Node values represent bootstrap percentages (only values above 50% are displayed). Each taxon is labeled with the virus name, the proposed group, or as unclassified. The star indicates the WfNgV1 placement in the tree. *Additional viruses, otherwise considered unclassified such as Indomegoura indica nege-like virus 1 and Astegopteryx formosana nege-like virus 1, are included in the Centivirus due to strong bootstrap support. The complete list of virus sequences along with their accession numbers is provided in S3 Table.

between the SP24 from WfNgV1 and its counterpart from members of the Sandewavirus group, as well as kita-like viruses.

## Discussion

In recent years, several nege/kitavirids have been reported not only from blood-sucking insects, but also from agriculturally important plant-feeding arthropods [12–14]. Understanding the evolutionary relationships among these viruses is pivotal as the emergence of current kitaviruses may represent recent cross-kingdom jumps from arthropod-infecting ancestors followed by new host adaptation processes, as documented, among others, for plant rhabdoviruses [45].

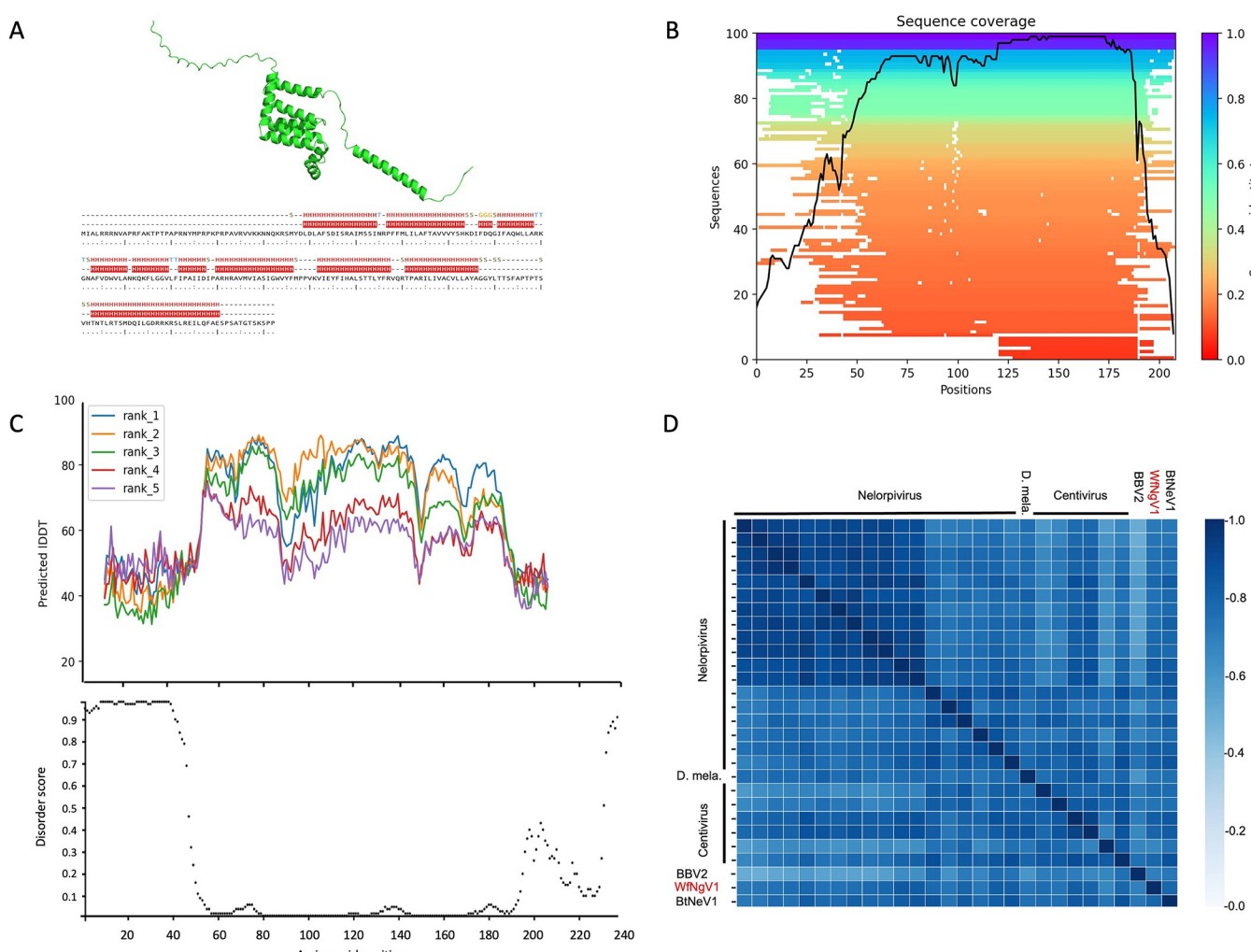

**Fig 4. Structural comparison of SP24 homologs.** A) Ribbon diagram of the putative SP24 of whitefly negevirus 1 (WfNgV1). B) Structure-based multiple sequence alignment across SP24 homologs. Heat-map representation of structure-based multiple sequence alignment (MSA). The color scale indicates the identity score scaled from 1 to 0 where 1 is full identity of sequences used in MSA to target sequence and 0 is the lowest identity with it. The black line indicates the relative coverage of SP24 from WfNgV1 with respect to the total number of aligned sequences by residues. C) Upper panel: Predicted local distance difference test (LDDT) plot depicting the quality assessment for five models (ranks) of WfNgV1 SP24. Scores above 70 (Y-axis) are considered reliable models. Lower panel: DISOPRED plot illustrating the mean index for predicted disorder. Residues at at both termini of the protein sequence (amino acid positions 1–60 and 190–240 show high disorder scores). D) Heat-map from TM-score values illustrating structural similarities across members of the Nelorpi- and Centivirus taxa, along with closest relatives. Abbreviations: D. mela: SP24 homolog from Drosophila melanogaster, BBV2: Beihai barnacle virus 2; WfNgV1: Whitefly negevirus 1; BtNeV1: Bemisia tabaci negevirus 1. Refer to S3 Table for the complete names and accession numbers corresponding to SP24 homologs used in this study.

Here, we present the complete genome of a new negevirus, provisionally designated whitefly negevirus 1 (WfNgV1), from the whitefly *Bemisia tabaci* MEAM1, a worldwide distributed plant-feeding insect [22]. Due to the distant homology observed for negevirus proteins [18], our analyses to infer the function and evolutionary relationships of WfNgV1 predicted proteins were mostly based at the amino acid level. WfNgV1 was found closely related to BtNeV1, a partially-sequenced virus discovered recently in whiteflies from Israel, and BtNeLV1-3, also partially sequenced from whiteflies in China [46]. The partial genome of BtNeV1(MW256675) denotes an organization similar to that of WfNgV1 (except for the missing 5' terminus of the polymerase coding region). For BtNeLV1, the sequence available from NCBI (MW256676) spans the complete coding regions for the polymerase (ORF1) and the glycoprotein (ORF2),

while for BtNeLV2-3 (MW256677 and MW256678) only the complete coding region for the polymerase (ORF1) is available.

Our phylogenetic analysis based on the RdRp revealed a close relationship between negevirids from B. tabaci and viruses in the nelorpi- and centivirus taxa. While the nelorpivirus group has long been proposed as a genus-level taxon, containing numerous documented homologs [7], the centivirus taxon, typified by Wuhan house centipede virus 1, was recently proposed and now includes several negevirids from distinct agriculturally important aphid species [13,14,42]. Furthermore, the proposed aphiglyvirus taxon, typified by Aphis glycines virus 3 [13], shows a distinct evolutionary lineage, being closer to sandewaviruses. However, the low bootstrap support for the node representing the clade including sandewa- and aphiglyviruses suggests the possibility of undiscovered homologs that could enhance sequence alignment quality, otherwise affected by conflicting evolutionary signals, and offer more precise insights into whether aphiglyviruses, centiviruses, and whitefly negeviruses constitute a single genus-level taxon.

Unlike previous studies, where phylogenetic analyses were focused exclusively on the RdRp, we analyzed extensively the SP24, which is conserved among nege- and kitaviruses. Results from our SP24 based phylogenetic analysis were consistent with those from the RdRp, where WfNgV1 and its closest relative BtNeV1, form a single clade with nelorpi- and centiviruses. Interestingly, the presumed SP24 from *Drosophila melanogaster* was found closely associated with centiviruses within a well-supported clade. This homolog was initially identified during a genome sequencing project for *D. melanogaster* conducted by the Berkeley Drosophila Genome Project in 2006 (https://www.ncbi.nlm.nih.gov/protein/ABC86319.1/). Its significance as an SP24 homolog became apparent during the characterization of the blueberry necrotic ring blotch virus, the type-member of the genus *Blunervirus* (*Kitaviridae*), where it was hypothesized to be a viral rather than a host protein [25]. Sequence based relationships among nelorpi-, centi-, the *D. melanogaster* SP24, and the whitefly negeviruses, WfNgV1 and BtNeV1, were supported by our structural-based analysis of the SP24, which resulted in high TM-scores. We contend that our analysis, incorporating a larger number of taxa, the concatenation of RdRp and SP24, and structural-based analysis of SP24, offers an accurate representation of the evolutionary relationships among these viruses, with whitefly negeviruses linking two previously proposed groups, nelorpi- and centiviruses.

## Supporting information

**S1 Fig. Percentage of contigs assembled from high throughput sequencing (HTS).** The pie chart shows the percent of RNA or DNA virus-related contigs associated to different hosts with respect to total contigs (n = 105) with homology to virus-encoded proteins identified by BLASTx.
(JPG)

**S1 Table. List of primers used for resequencing the whitefly negevirus 1 (WfNgV1) genome.** The following PCR parameters are recommended for all the primer sets used to resequence the genome: 96˚C x 4 min, 40 cycles of 96˚C x 1 min, 55˚C x 30 sec, and 72˚C x 1.5 min, a final additional extension at 72˚C x 5 min. For rapid-amplification of cDNA-ends (RACE) refer to manufacturer instructions.
(DOCX)

**S2 Table. Virus names and accession numbers for amino acid sequences used in the alignments for Fig 2.** n.a. denotes no homologue available.
(DOCX)

**S3 Table. Virus names and accession numbers for amino acid sequences corresponding to putative structural protein of 24 kDa (SP24) and RNA-dependent-RNA-polymerase (RdRp) used for phylogenetic analyses (Fig 3).** n.a. denotes no homologue available. (DOCX)

## Acknowledgments

Authors would like to acknowledge papaya growers in Santa Elena for granting access to their fields and the Department of Biodiversity of the Ecuadorean Ministry of the Environment for granting *Genetic Resource Access Permit* # MAE–DNB–CM–2018–0098.

## Author Contributions

**Conceptualization:** Diego F. Quito-Avila, Francisco F. Flores.

**Formal analysis:** Edison Reyes-Proaño, Gerardo Armijos-Capa, Ricardo I. Alcalá Briseño, Robert Alvarez, Francisco F. Flores.

**Methodology:** Diego F. Quito-Avila, Edison Reyes-Proaño, Gerardo Armijos-Capa, Ricardo I. Alcalá Briseño, Robert Alvarez, Francisco F. Flores.

**Software:** Edison Reyes-Proaño, Gerardo Armijos-Capa, Ricardo I. Alcalá Briseño, Robert Alvarez.

**Supervision:** Diego F. Quito-Avila.

**Writing – original draft:** Diego F. Quito-Avila.

**Writing – review & editing:** Diego F. Quito-Avila, Edison Reyes-Proaño, Gerardo Armijos-Capa, Robert Alvarez, Francisco F. Flores.

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
