## [Decision Letter · Decision Letter 0]

7 Mar 2024

PONE-D-24-05885

Analysis of a new negevirus-like sequence from Bemisia tabaci unveils a potential new taxon linking nelorpi- and centiviruses

PLOS ONE

Dear Dr. Quito-Avila,

Thank you for submitting your manuscript to PLOS ONE. After careful consideration, we feel that it has merit but does not fully meet PLOS ONE’s publication criteria as it currently stands. Therefore, we invite you to submit a revised version of the manuscript that addresses the points raised during the review process.

We look forward to receiving your revised manuscript.

Kind regards,

Rhys Harold Parry

Academic Editor

PLOS ONE

Journal Requirements:

2. ""In your Methods section, please provide additional information regarding the permits you obtained for the work. Please ensure you have included the full name of the authority that approved the field site access and, if no permits were required, a brief statement explaining why."""

Additional Editor Comments:

Dear Dr Quito-Avila,

Your manuscript has now been reviewed by two experts in the field. They both find the results of interest and recommend publication pending several minor modifications.

Please meet or respond to the concerns raised in an outlined review document and submit a resubmission of your manuscript.

Dr. Rhys Parry

Reviewers' comments:

Reviewer's Responses to Questions

**Comments to the Author**

1. Is the manuscript technically sound, and do the data support the conclusions?

Reviewer #1: Yes

Reviewer #2: Yes

2. Has the statistical analysis been performed appropriately and rigorously? 

Reviewer #1: N/A

Reviewer #2: Yes

3. Have the authors made all data underlying the findings in their manuscript fully available?

Reviewer #1: Yes

Reviewer #2: Yes

4. Is the manuscript presented in an intelligible fashion and written in standard English?

Reviewer #1: Yes

Reviewer #2: Yes

5. Review Comments to the Author

Reviewer #1: Analysis of a new negevirus-like sequence from Bemisia tabaci unveils a potential new taxon linking nelorpi- and centiviruses

The paper by Quito-Avila and colleagues describes the identification of a novel nege-like virus (complete genome sequenced) in the whitefly Bemisia tabaci, an important agricultural pest. Viruses found in this species are of importance given the hosts agricultural association and evidence suggesting that insect specific viruses such as the nege-like viruses may inhibit alphavirus infection offering a potential method of controlling arthropod-borne pathogens. The authors provide an in-depth characterisation of this novel virus including phylogenetic and structural analyses of structural proteins. Overall, I have no major concerns about this study, the manuscript is well written, and the methods used are adequate. However, the following should be addressed before this manuscript is suitable for publication in PLoS One.

Figure 3:

The vertical gaps between branches are inconsistent, have tips been manually pruned from the trees or is this an artefact from collapsing branches? Furthermore, the scale bar across the trees is quite large. Please add to the figure legend what the scale bar represents (amino acid substitutions per site?). Please provide further information about the alignments - length, pairwise identity, etc- used to infer these trees. If possible, alignments should be made publicly available in a data repository like Github to enable reproducibility. Likewise, it would be great if structure of SP24 was made available.

Methods:

- Please clarify whether the RNA from the ten whiteflies was extracted individually before being pooled to form a single library.

- The authors state that they filtered out host-associated contigs by performing BLASTn against the Bemisia tabaci genome. The exclusion criteria here is unclear, please clarify whether all hits to the B. tabaci genome excluded and the e-value cutoff used. Similarly, e-value should be specified when describing the BLASTx analysis, i.e., "Homology of remaining contigs was explored by BLASTx using the non-redundant protein database (as of August 2023)."

- Minor point, the authors include versioning and citations for some software in the methods section but not for others. Please include this information for all software used.

- Please describe how the primers used in the PCR were designed (e.g., based on the whitefly negevirus 1 sequence or on some other sequence).

- Please specify how bootstrap support was calculated.

Figure legends:

- In Figure 4B, please clarify what is meant by largest and lowest identity. This should also be included as a label on the figure key.

- S1 Fig. I don’t see any information on how contig abundance is calculated. If this is just the percentage of contigs assigned to each group, I would refer to it as such, rather than using the term "abundance." Regardless, please describe the method for you used for calculating this metric.

- Panel D) seems to be missing in the Figure 2 caption text.

Minor comments:

- In the abstract the authors state that “Both viruses form a clade sharing a recent ancestor with the proposed nelorpivirus and centivirus taxa”. From my reading of the text, the authors are referring the most recent common ancestor of these two taxa. This should be specified as the current wording implies that these taxa diverged recently, a conclusion that is not supported by the data.

- Line 122: “Most accessions selected through PSI-BLAST encompassed both RdRp and SP24.” This current wording makes it seem that RdRp and SP24 were found on the same sequence, whereas I assume the intended meaning is the following: For most nege/kitavirids identified through PSI-BLAST both the RdRp and SP24 sequences were found.

- Line 167: Phyton 3 should be Python 3

- Throughout the manuscript and figures capitalisation and italicisation of viral groups is inconsistent. Please refer to the following guide: https://ictv.global/faqs

Reviewer #2: Whitefly negevirus 1 is a fascinating novel virus identified in an agriculturally significant host and falls within an evolutionarily interesting clade. The authors describe this novel virus in detail genomically and phylogenetically. The phylogenies in this paper also appear to link Nelorpivirus and Centivirus, hopefully bringing us a step to formally ratifying these clades. The methodology is reproducible and well-described overall. I have generally minor comments and queries throughout the manuscript as well as some suggestions for figure edits that should improve the ease of interpretation for a reader. These are detailed in a line-by-line review uploaded as an attachment. Overall, this is a robust paper that is sure to be of interest to the readers of PLOS ONE, and will be particularly valuable to readers focused on arthropod-associated viruses or unique and divergent RNA virus clades.

6. PLOS authors have the option to publish the peer review history of their article (what does this mean?). If published, this will include your full peer review and any attached files.

Reviewer #1: No

Reviewer #2: No

---

## [Author Response · Author response to Decision Letter 0]

11 Apr 2024

Authors’ Response to Review Comments

Dear Editor,

Please find below the responses to each comment/suggestion diligently made by each reviewer.

As part of the “response to reviewers” step, we are submitting a change-track version of the revised manuscript, so that reviewers and you can confirm our amendments.

We want to thank both reviewers for such nice words about our work, but specially for taking the time to read the manuscript thoroughly making pertinent suggestions and comments.

Reviewer #1: Analysis of a new negevirus-like sequence from Bemisia tabaci unveils a potential new taxon linking nelorpi- and centiviruses

The paper by Quito-Avila and colleagues describes the identification of a novel nege-like virus (complete genome sequenced) in the whitefly Bemisia tabaci, an important agricultural pest. Viruses found in this species are of importance given the hosts agricultural association and evidence suggesting that insect specific viruses such as the nege-like viruses may inhibit alphavirus infection offering a potential method of controlling arthropod-borne pathogens. The authors provide an in-depth characterisation of this novel virus including phylogenetic and structural analyses of structural proteins. Overall, I have no major concerns about this study, the manuscript is well written, and the methods used are adequate. However, the following should be addressed before this manuscript is suitable for publication in PLoS One.

Figure 3:

The vertical gaps between branches are inconsistent, have tips been manually pruned from the trees or is this an artefact from collapsing branches? Furthermore, the scale bar across the trees is quite large. Please add to the figure legend what the scale bar represents (amino acid substitutions per site?). Please provide further information about the alignments - length, pairwise identity, etc- used to infer these trees. If possible, alignments should be made publicly available in a data repository like Github to enable reproducibility. Likewise, it would be great if structure of SP24 was made available.

Response:

The spaces between branches have been fixed as recommended. The phylogenetic tree was based on amino acid sequences. So, this information has been added to the figure legend. The amino acid sequence alignments used for the phylogenetic analysis, as well as the 3D structure of SP24 (.pdb) will be made available upon request to the corresponding author.

Methods:

- Please clarify whether the RNA from the ten whiteflies was extracted individually before being pooled to form a single library.

Response:

This part has been clarified as suggested. (See related comment for reviewer 2). 

- The authors state that they filtered out host-associated contigs by performing BLASTn against the Bemisia tabaci genome. The exclusion criteria here is unclear, please clarify whether all hits to the B. tabaci genome excluded and the e-value cutoff used. Similarly, e-value should be specified when describing the BLASTx analysis, i.e., "Homology of remaining contigs was explored by BLASTx using the non-redundant protein database (as of August 2023)."

Response:

Yes. We have clarified this part by adding more information about the host genome filtered.

- Minor point, the authors include versioning and citations for some software in the methods section but not for others. Please include this information for all software used.

Response:

We have checked throughout the manuscript and added software versions accordingly.

- Please describe how the primers used in the PCR were designed (e.g., based on the whitefly negevirus 1 sequence or on some other sequence).

Response:

Yes. We recognize that this part was not clear. We have added a few lines to make it clear.

- Please specify how bootstrap support was calculated.

Response:

Details on the bootstrap support, and the corresponding citation, have been added.

Figure legends:

- In Figure 4B, please clarify what is meant by largest and lowest identity. This should also be included as a label on the figure key.

Response:

The text in the figure legend has been modified for clarification.

- S1 Fig. I don’t see any information on how contig abundance is calculated. If this is just the percentage of contigs assigned to each group, I would refer to it as such, rather than using the term "abundance." Regardless, please describe the method for you used for calculating this metric.

Response:

We agree. Modifications to the text and the pie chart have been made, in response to both reviewers.

- Panel D) seems to be missing in the Figure 2 caption text.

Response:

Yes. Thank you for catching this mistake. The corresponding text has been added.

Minor comments:

- In the abstract the authors state that “Both viruses form a clade sharing a recent ancestor with the proposed nelorpivirus and centivirus taxa”. From my reading of the text, the authors are referring the most recent common ancestor of these two taxa. This should be specified as the current wording implies that these taxa diverged recently, a conclusion that is not supported by the data.

Response:

Yes. We agree. We have accepted this suggestion and reworded the text accordingly.

- Line 122: “Most accessions selected through PSI-BLAST encompassed both RdRp and SP24.” This current wording makes it seem that RdRp and SP24 were found on the same sequence, whereas I assume the intended meaning is the following: For most nege/kitavirids identified through PSI-BLAST both the RdRp and SP24 sequences were found.

Response:

Yes. The way the reviewer suggests is exactly what was meant. Hence, the text was replaced accordingly.

- Line 167: Phyton 3 should be Python 3

Response:

This has been corrected.

- Throughout the manuscript and figures capitalisation and italicisation of viral groups is inconsistent. Please refer to the following guide: https://ictv.global/faqs

Response:

We have attempted to homogenize the virus name notation according to ICTV guidelines. However, it should be note that most viruses mentioned in this work, are actually informal names, as negeviruses have not yet received formal taxonomical assignments by the ICTV. In other words, these are not formally considered “species”. But if needed, we will be ready to work on this during the editorial stage.

Reviewer #2: Whitefly negevirus 1 is a fascinating novel virus identified in an agriculturally significant host and falls within an evolutionarily interesting clade. The authors describe this novel virus in detail genomically and phylogenetically. The phylogenies in this paper also appear to link Nelorpivirus and Centivirus, hopefully bringing us a step to formally ratifying these clades. The methodology is reproducible and well-described overall. I have generally minor comments and queries throughout the manuscript as well as some suggestions for figure edits that should improve the ease of interpretation for a reader. These are detailed in a line-by-line review uploaded as an attachment. Overall, this is a robust paper that is sure to be of interest to the readers of PLOS ONE, and will be particularly valuable to readers focused on arthropod-associated viruses or unique and divergent RNA virus clades.

Comments

1. Line 93: The data set is comprised of one library constructed from ten individual whiteflies. It would be helpful to explain why the samples were combined and to indicate at which stage samples were combined (i.e., before or after RNA extraction).

Response:

As there was a similar comment from reviewer 1, we added a few lines to address this part. In short, in preliminary extractions we realized that the amount and quality of RNA did not meet the requirements for HTS. Hence, we pooled 10 whiteflies and performed the extraction as a single preparation.

2. Line 132: It is unclear how many ambiguously aligned positions were removed. Can the authors provide the lengths of the original alignments and compare to the lengths of trimmed alignments?

Response:

This information has been added.

3. Line 174: What were the percentage identity cut-offs for considering a contig to be viral? Were these contigs further analysed (phylogenetically for example) to confirm they were indeed viral? And how were host organisms assigned? This would greatly enhance the level of detail in the methods section.

Response:

We added information (in the Genome assembly and analysis section) as to how we performed the BLASTx analysis and how a “virus-related” contig was regarded as such. We did not perform further phylogenetic analyses for confirming BLASTx results, as our focus was the negevirus-like that we report in this study.

4. Line 174: Does this data set of 105 contigs include RNA viruses or all viruses? This should be stated in the methods as well as the figure legend for Supplementary Figure 1.

Response:

While plant-, fungi- and insect- related viruses were all RNA viruses, bacteriophage-sequences were mostly from DNA viruses. We agree. This should have been clarified in the text and legend of S1 Fig. (Fixed now).

5. Figure 2 legend: A description of Panel D appears to be missing in this figure legend.

Response:

Yes. Thank you. We added the missing description.

6. Line 257: Please refer to the relevant figures earlier in each section for ease of understanding.

Response:

We have adhered to the journal guidelines for calling Figures in the text. But, we’ll be ready to fix these issued at the editorial stage if necessary.

7. Figure 2B: The highlighted red valine residue and lack of position markings seems to imply the fourth conserved cysteine residue is not present and is instead a conserved valine residue. Consider marking the positions of interest and highlighting the fourth cysteine residues in position 114, perhaps by shading the column in grey as it is not universally conserved like positions 70, 84, and 90.

Response:

Thanks for the suggestion. Agree. We have modified the figure accordingly. Additional slight modifications have been done to fig 2 to comply with suggestions regarding potential readers with color blindness. (Good suggestion!!).

8. Line 258: What are the “additional conserved residues”? Cysteine?

Response:

The text has been modified to clarify this observation.

9. Line 265: This section needs more references to the relevant figure, especially earlier on. It’s very difficult to follow the text without knowing which figure represents the described data.

Response:

This section, which reads “ORF 2 deduced 49 kDa protein in WfNgV1 exhibits motifs shared among chropara- and negevirus glycoproteins” tells the reader what the paragraph is about, and the reader is only referred to Fig 2B (nothing else). Please keep in mind that Figure 2 has 4 panels (2A for the RdRp, 2B for the glycoprotein, 2C for the SP24, and 2D for that additional ORF 4 found in WfNgV1 and closest relative).

Perhaps the confusion arises from the fact that the whole figure description (legend) is before. But that is what the journal format calls for. Nevertheless, the amendments made to Fig 2B and the text (previously suggested by Reviewer 2), will make the whole paragraph clearer.

10. Line 267: “(refer to the subsequent section)” This is confusing as it is unclear when reading what subsequent section is being referred to.

Response:

We have reworded that line to avoid confusion.

11. Figure 2C: It may be helpful to highlight the exact residues considered to be conserved here. Perhaps they can be bolded so it is clear, particularly in some of the less obviously conserved positions (e.g., in the first shaded column of Figure 2C, bold each occurrence of the letter D).

Response:

We accepted this suggestion and applied it to panel 2B as well for uniformity.

12. Lines 271-289: The entire section refers often to the amino acid compositions of different parts of the WfNgV1 genome. The significance of these observations are not discussed. I encourage the authors to expand on this idea in both the results and discussion sections so it is clear why the amino acid compositions matter.

Response: 

We have added a few lines to the discussion section to address this observation.

13. Line 285: It may be helpful to put the single-letter code in brackets when a full amino acid name is written to assist those who do not have the abbreviations memorized. That will make it easier to quickly refer to the figure and identify the residue in question.

Response:

We have addressed this issue by adding single letter codes only for those amino acids that will be used again in following lines/paragraphs.

14. Figure 3A: It is interesting that the Kitaviridae and kita-like RdRp sequences in this tree are not monophyletic, in contrast to the other trees. What do the authors think of this observation?

Response:

It is well known, that the RdRp is one of the most conserved, but also ancient, proteins in RNA viruses. As such, it is not unusual for RdRps of some viruses to have undergone recombination events which can influence their evolutionary pathways. In plants, viral mixed infections —involving related or unrelated viruses— are rather common, hence plant viruses are particularly susceptible to recombination and/or reassortment events, especially for those with segmented genomes, as is the case for Bluner/Cile/Higre (members of the Kitaviridae). The observation made by the reviewer aligns with others (including ours), which altogether may even suggest that eventually the Kitaviridae might be split into two distinct families (one that encompasses blunerviruses and other for Cile/Higre-viruses, which are notably closest to each other). Having said that, the present work does not aim to investigate any further into the evolution of kitaviruses per se, but to share a new nege- (insect-associated) virus and look at its most recent evolution. We thank this reviewer for prompting this interesting subject though!

15. Figure 3: The spacing in between certain branches is strange. Is this simply a visual artifact or are these large gaps intentionally added to signify something?

Response:

Indeed. It is the result of manually collapsing branches in order to increase the main focus of our figures, which is the newly discovered virus. But we accept the suggestion and have modified the figure to make the spacing uniform.

16. Figure 3: What do the branch lengths and scale bar represent? Number of amino acid changes per site? This information should be added to the figure legend.

Response:

Yes. We added the suggested information to the figure legend.

17. Figure 4: The authors may want to reconsider the colours used in this figure (particularly panels C and D). The current palettes may be difficult for readers with reduced colour vision or those reading in black and white.

Response:

Thank you. We have modified panel D in this figure, as suggested. We would like to keep the current version of panel C, as we consider that this part is easy to interpret even if a person cannot distinguish those colors.

18. Figure 4D: As the scale is from 0 to 1, it may be better to have a single-colour scale, ie, eliminate the blue shades and represent 0 with white and 1 with dark red.

Response:

This has been modified.

19. Line 380: Avoid referencing figures in the discussion, that should be restricted to results. Here, the results should be expanded on in the context of currently available literature.

Response:

Duly noted. We have deleted fig calls in the discussion section.

20. Line 385: Low bootstrap support indicates low confidence in the relationships inferred from presented topology; other causes, such as poor alignment and conflicting evolutionary signals, must be acknowledged.

Response:

Thank you for this valuable observation. We have incorporated a few lines, using your suggestion, to clarify that aspect.

21. Line 569: I highly recommend representing these data differently, at least by making it a flat pie chart rather than 3D. Also, the colours are unfortunately completely indistinguishable in black and white and quite difficult to distinguish for those with colour blindeness. Please change the colour palette or consider labelling the slices directly with the relevant category.

Response:

These are details that authors usually do

---

## [Editor Report · Decision Letter 1]

2 May 2024

Analysis of a new negevirus-like sequence from Bemisia tabaci unveils a potential new taxon linking nelorpi- and centiviruses

PONE-D-24-05885R1

Dear Dr. Quito-Avila,

We’re pleased to inform you that your manuscript has been judged scientifically suitable for publication and will be formally accepted for publication once it meets all outstanding technical requirements.

Kind regards,

Rhys Harold Parry

Academic Editor

PLOS ONE

---

## [Editor Report · Acceptance letter]

7 May 2024

PONE-D-24-05885R1 

PLOS ONE

Dear Dr. Quito-Avila, 

I'm pleased to inform you that your manuscript has been deemed suitable for publication in PLOS ONE. Congratulations! Your manuscript is now being handed over to our production team.

Kind regards, 

on behalf of

Dr. Rhys Harold Parry 

Academic Editor

PLOS ONE